# Multi-Step Sequence Flood Forecasting Based on MSBP Model

**Yue Zhang** [1], **Juanhui Ren** [2], **Rui Wang** [1], **Feiteng Fang** [1] and **Wen Zheng** [1,3,*]

[1] Institute of Public-Safety and Big Data, College of Data Science, Taiyuan University of Technology, Taiyuan 030060, China; zhangyue0990@link.tyut.edu.cn (Y.Z.); wangrui1070@link.tyut.edu.cn (R.W.); fangfeiteng7872@link.tyut.edu.cn (F.F.)

[2] College of Data Science, Taiyuan University of Technology, Taiyuan 030060, China; chunyanxueyan@163.com

[3] Center for Big Data Research in Health, Changzhi Medical College, Changzhi 046000, China

[*] Correspondence: zhengwen@tyut.edu.cn

**Abstract:** Establishing a model predicting river flow can effectively reduce huge losses caused by floods. This paper proposes a multi-step time series forecasting model based on multiple input and multiple output strategies, and this model is applied to the flood forecasting process of a river basin in Shanxi, which effectively improves the engineering application value of the flood forecasting model based on deep learning. The experimental results show that after considering the seasonal characteristics of the river channel and screening the influencing factors, a simple neural network model can accurately predict the peak value, the peak time and flood trends. On this basis, we proposed the MSBP (Multi-step Back Propagation) model, which can accurately predict the flow trend of the river basin 20 h in advance, and the NSE (Nash Efficiency) is 0.89. The MSBP model can improve the reliability of flood forecasting and increase the internal interpretability of the model, which is of great significance for effectively improving the effect of flood forecasting.

**Keywords:** MSBP model; flood forecasting; multi-step forecast

## 1. Introduction

Effective flood prevention is an important part of water resources planning [1]. At the same time, floods are also the most common natural disasters in the world, especially in areas dominated by monsoon climates, where floods occur more frequently and are mostly concentrated in the rainy season [2,3]. Flood forecasting is one of the most important non-structural measures for flood control [4]. The establishment of an early warning system can effectively reduce flood damage [5–7]. At present, flood forecasting is mainly divided into two categories. One is the method based on hydrological knowledge: a large amount of professional knowledge is required as a basis; the other is a data-driven method [8]: based on the existing data in the basin, machine learning algorithms are used to mine data relationships for prediction. In traditional hydrological knowledge-based forecasting methods, data acquisition is mostly manual collection, and the storage is messy, resulting in a lack of hydrological data, and even difficult to obtain data in some remote areas. The data-driven artificial neural network only needs to input historical hydrological data in the basin [9,10], and it can complete flood forecasting through a self-learning function [11], avoiding the complicated modeling process, cumbersome parameter selection and long forecasting time in traditional hydrological methods. Therefore, it is favored by more and more researchers. Through the application of the genetic algorithm, particle swarm algorithm and other methods, it is found that machine learning can accurately capture important information such as flood peak flow value and peak present time [12], which can effectively improve the forecasting effect [13–16].

The artificial neural network can predict the flow of a certain location based on the flow of the upstream position of the river basin [17], which has a wider application range and stronger generalization ability than traditional methods [18]. Especially in areas with sufficient and more accurate data, the use of the BP (Back Propagation) neural network

model can significantly improve the accuracy of flood forecasting [19]. BP neural network can overcome the shortcomings of traditional hydrological knowledge-based models where the convergence speed is slow and the extreme value is easily reached [20], so it is widely used. Researchers continue to make improvements to the BP neural network [21–23], combining the characteristics of the current watershed with the BP neural network, improving the input data of the BP neural network, optimizing the hidden layer activation function, and improving the flood forecasting effect. Because GA (Genetic Algorithm) has good simulation accuracy and can obtain the optimal solution in the global scope, the researchers combined GA and BP models [24,25] to effectively improve the prediction effect in the basin.

On the other hand, the use of time series analysis technology can make rapid and effective forecasts based on past data and improve the accuracy of flood forecasting [26,27], even if there is only a weak-to-medium relationship between the predicted value and the observed value. At present, in different watersheds, researchers have explored many different multi-step forecasting methods and have achieved great results. In the Three Gorges Reservoir basin, Yanlai Zhou et al. used UKF (Unscented Kalman Filter) combined with artificial neural network [28], genetic algorithm and least-squares methods [19] to capture the relationship between rainfall and runoff and reduce the uncertainty of flood forecasting. Fi-John Chang et al. [29] used a static neural network and two dynamic neural networks to screen and predict in the Taiwan watershed. Among them, the dynamic network effectively finds the dependence between factors by realizing multi-step long-term prediction. At the same time, the latest observations and output values are used to repeatedly adjust the model parameters, which improves the accuracy of multi-scale analysis and effectively alleviates the problem of time lag [30]. In the case of considering the weight of time and space [31], linking rainfall and runoff series can increase the internal interpretability of the model [32]. Although the LSTM (Long-Short Term Memory) model can simulate the flood process well, it is difficult to achieve the effect of real-time forecasting [33], and the forecasting time is long. Unlike single-step forecasting, multi-step forecasting usually faces uncertainties from various sources. For example, the accumulation of errors and lack of information make multi-step prediction more difficult [28].

In summary, although past research has clarified the predictive ability of BP neural network, few people have achieved its multi-step advanced time series prediction. This paper proposes a multi-step time series forecasting model based on multiple input and multiple output strategies, the MSBP model (Multi-Step BP), and verifies the feasibility of the model in the actual watershed. This paper first selects BP neural network and random forest models to verify the feasibility of data, machine learning methods, and data processing in the current watershed in the Fenhe Reservoir, Zhaishang area of Shanxi Province, China. The above method can accurately capture the peak discharge value and peak present time, and it obtains the Nash coefficient of 0.9–0.99. Then, by applying the MSBP model in the basin, it can achieve a multi-step sequence forecast N hours in advance and accurately predict the future flow trend.

## 2. Application

### 2.1. Background of the Study Area

As shown in Figure 1, the study area is located in the upper reaches of the Fenhe Reservoir in Shanxi Province, China, in the northwest corner of Taiyuan City. The main river length above the section is 34.2 km, and the interval control area is 1551 km$^2$. This section is joined by tributaries such as Lion River, Tianchi River, Tunlan River, Yuanping River and Dachuan River. The geological environment in the basin is complex, including 358.66 km$^2$ of metamorphic rock shrubland, accounting for 23.13% of the catchment area, limestone shrubland of 224.15 km$^2$, accounting for 14.45% of the catchment area, and sand shale shrubland of 350.13 km$^2$, accounting for Jiji. The water area is 22.57%; the sandy shale forest mountain area is 69.16 km$^2$, accounting for 4.46% of the catchment area; the sandy shale rocky mountain area is 548.90 km$^2$, accounting for 35.39% of the catchment area.

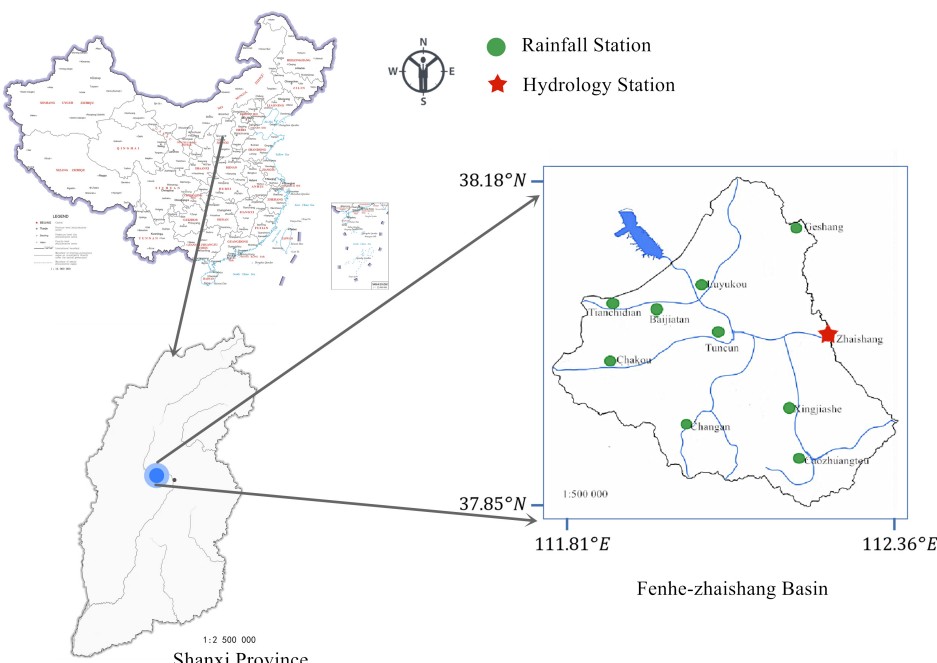

**Figure 1.** Map of major rainfall sites. After characteristic analysis, the above 10 rainfall stations are the main influencing factors in the flood forecast of Zhaishang. The green dot is the rainfall station, and the red dot is the forecast point of Zhaishang flow.

The Fenhe Reservoir, Zhaishang section is mainly controlled by the mainstream of the Fenhe River. The average longitudinal slope of the river is 3.39%, the roughness of the river bed is about 0.02, and the average width of the basin is 40.6 km. The multi-year average annual precipitation of the interval basin was 437.5 mm, the maximum annual precipitation was 642.8 mm, which occurred in 1988; and the minimum annual precipitation was 172.1mm, which occurred in 1972; the multi-year average flow was 11.2 m$^3$/s, and the maximum instantaneous flow was 2130 m$^3$/s, on 22 August 1967. In the Fen River Basin, due to the seasonality of the river, the rainy season is concentrated in June to August each year. Most of the floods occur from the end of July to mid-August each year.

There are 4 water level stations (Gaosheng, Malan, Xingjiashe, Jialequan), 1 hydrological station (Zhaishang), and 58 rainfall stations (basic stations, small and medium river stations, flash flood stations) in Fenhe Reservoir, Zhaishang section.

### 2.2. Data Collection

At present, there are 58 rainfall stations in the Fenhe Zhaishang Basin. However, some of the rainfall stations were constructed later, resulting in incomplete data in the basin. Direct input to the model will reduce the accuracy of the model. As shown in Figure 2, this paper analyzes the correlation between rainfall stations and Zhaishang flow stations, and 10 rainfall stations with much higher correlation (correlation greater than 0.1) than other stations were selected as the basic data of the model, reducing the input sequence model and reducing the data complexity. The site distribution of the final input model is shown in Figure 1.

In addition, the cumulative i-h catchment average rainfall of the flood (i = 1, 2, 3) and the flow of j hours ahead (j = 1, 2, 3) are used as the primary predictors (independent variables), and the period-by-period flood flow is taken as the forecast object (dependent variable), the forecast factor is screened through statistical analysis. The predictors screened by the correlation coefficient as the objective function are:

(1)     accumulate 2 h basin area average rainfall;
(2)     accumulate 3 h basin area average rainfall.

The processed data have a stronger correlation with the flow of the Zhaishang station, which is conducive to better training data for the model, quickly obtaining the optimal bias term, and performing flood forecasting.

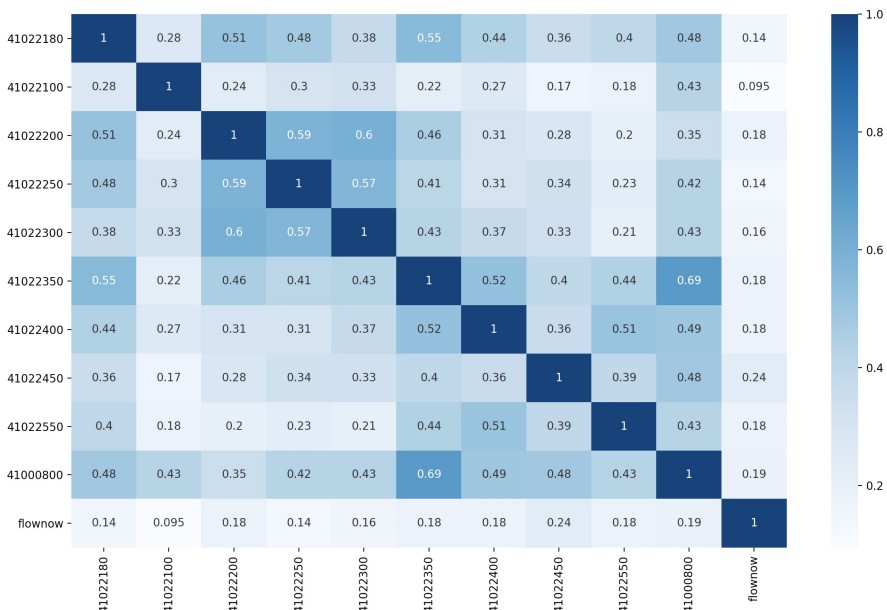

**Figure 2.** Correlation coefficient matrix of predictors. The figure shows that the correlation of the predictors used in this model is strongly related to the flow of Zhaishang. 41022180 is the code of Luyukou rainfall station, 41022100 is the code of Geshang rainfall station, 41022200 is the code of Tianchidian rainfall station, 41022250 is the code of Baijiatan rainfall station, 41022300 is the code of Chakou rainfall station, 41022350 is Tuncun The code of the rainfall station, 41022400 is the code of Chang'an rainfall station, 41022450 is the code of Caozhuangtou rainfall station, 41022550 is the code of Xingjiashe rainfall station, 41000800 is the code of Zhaishang rainfall station. Flownow is the river flow at the Zhaishang site.

In the Fenhe Reservoir, Zhaishang Interval Basin, complete and prominent flood peak data have been integrated into 11 flood data points from 1994 to 2006 and 2016 to 2018, which is No. 19940806, No. 19960803, No. 19990827, No. 20010810, No. 20060903, No. 20080814, No. 20160719, No. 20160824, No. 20170723, No20170727 and No. 20180713 that are named as the time of occurrence. After data preprocessing and integration, the data time step is 1h, including the data before the flood peak, when the flood peak appears, and after the flood peak. Among them, 8 single peaks and 1 compound flood were selected for model calibration, and 1 single peak and 1 compound flood were used for model verification. We used the BP neural network model and random forest to verify the 11 floods to ensure the practicability of the data in this paper. The results are shown in Tables 2 and 3 . The first column of the table is the flood number named by the time of occurrence. At the same time, the MSBP model was used to predict 11 floods in multi-step time series, and the results of representative single-peak flood No. 19940806 and compound flood peak flood No. 20170727 were displayed.

The data selected strictly abide by the requirements of the Hydrological Information Forecast Specification (GB/T 22482-2008). The historical flood data, especially the flood data since the 1990s, are selected when establishing forecasting plans (calibration or verification). All hydrological stations and rainfall stations used in this project are the national basic station network. The data have been reviewed by the Taiyuan Hydrological Branch, accepted by the provincial bureau, and compiled by the river basin. The authenticity of data is very reliable.

## 3. Materials and Methods

*3.1. BP Neural Network*

BP neural network is a "universal model and error correction function", each time the error analysis is carried out according to the training results and the expected results, and then the weights and thresholds are modified, step by step, the output is consistent with the expected results.

BP (Back-propagation) neural network is one of the most effective multi-layer neural network learning methods. It is characterized by forward transmission of signals and backward propagation of errors. By continuously adjusting the weight value of the network matrix, the final output of the network is as good as the expected output. It may be close to achieve the purpose of training and prediction.

BP neural network usually contains three layers of structure, namely input layer, hidden layer and output layer. There are two stages in the construction of a BP neural network: the first stage is the forward propagation of the signal. In this stage, our data are received by an input layer and processed by a hidden layer. Then, the output layer will receive the processed data from the hidden layer, giving us the final prediction results. Finally according to the prediction result and the true value , we can form the cost function. The second stage is the back propagation of errors. Using the gradient descent method [28], the weights of the output layer, hidden layer and the input layer can be adjusted in turn. In the process of back-propagation, the value of each parameter is continuously adjusted and optimized according to the error; the process is repeated continuously until convergence.

Refer to Algorithm 1 for the specific update process.

---

**Algorithm 1:** BP neural network algorithm.

---

**Input:** training set $D = (x_k, y_k)_{k-1}^{m}$;Learning rate $\eta$

**Output:**

1:**function** BP(D,$\eta$)

2:  Randomly initialize all connection weights and
     thresholds in the network within the range of (0,1)

3:  **repeat**

4:    for all $(x_k, y_k) \in D$ do

5:      Calculate the current sample output
          $\hat{y}_k = f(\beta_j - \theta_j)$

6:      Calculate the gradient of the output neuron
          $$g_j = -\frac{\partial E_k}{\partial \hat{y}^k} \cdot \frac{\partial \hat{y}^k}{\partial \beta_j}$$
          $$= -\left(\hat{y}_j^k - y_j^k\right) f'(\beta_j - \theta_j)$$
          $$= \hat{y}_j^k \left(1 - \hat{y}_j^k\right)\left(\hat{y}_j^k - y_j^k\right)$$

7:      Calculate the neuron gradient term of the hidden
          layer $e_h = -\frac{\partial E_k}{\partial b_h} \cdot \frac{\partial b_h}{\alpha_h}$
          $$= -\sum_{j=1}^{l} \frac{\partial E_k}{\partial \beta_j} \cdot \frac{\partial \beta_j}{b_h} f'(\alpha_h - \gamma_h)$$
          $$= \sum_{j=1}^{l} w_{hj} g_j f'(\alpha_h - \gamma_h)$$
          $$= b_h(1 - b_h) \sum_{j=1}^{l} w_{hj} g_j$$

8:        Update weight $\triangle w_{hk} = \eta g_j b_h, \triangle v_{hk} = \eta e_j x_i$

9:        Update threshold

10:     **end for**

11:   **until** Stop condition reached

12:**end function**

---

Given the training set $D = \{(x_1, y_1), (x_2, y_2), \ldots, (x_m, y_m)\}, x_i \in \mathbb{R}^d, y_i \in \mathbb{R}^l$, the input is described by d attributes, and the output is an l-dimensional value vector. For the convenience of discussion, a multilayer feedforward network structure with d neurons, l

output neurons and q hidden layer neurons is given. The threshold of the j neuron in the output layer is represented by $\theta_j$, and the threshold of the h neuron in the hidden layer is represented by $\gamma_h$. The connection weight between the i neuron in the input layer and the h neuron in the hidden layer is $v_{ih}$, and the connection weight between the h neuron in the hidden layer and the j neuron in the output layer is $w_{ih}$. Remember the input received by the h neuron in the hidden layer as $\alpha_h = \sum_{i=1}^{d} v_{ih}x_i$. The input and output received by the j neuron in the output layer is $\beta_j = \sum_{h=1}^{q} w_{hj}b_h$, where $b_h$ is the output of the h neuron in the hidden layer. For the training set $D = (x_k, y_k)$, it is supposed the output of the neural network is $\hat{y}_k = \left(\hat{y}_1^k, \hat{y}_2^k, \ldots, \hat{y}_l^k\right)$, which is $\hat{y}_k = f(\beta_j - \theta_j)$. The mean square error of the network on $(x_k, y_k)$ is $E_k = \frac{1}{2}\sum_{j=1}^{l}\left(\hat{y}_j^k - y_j^k\right)^2$. The gradient of the neurons in the output layer is denoted as $g_i$, the neuron gradient of the hidden layer is denoted as $e_h$. The BP neural network learning rate is $\eta$.

### 3.2. Random Forest

Random forest [34] is an important ensemble learning method based on Bagging (Bootstrap aggregating), which can be used for classification, regression and other problems. It uses the bootstrap re-sampling method to extract multiple samples from the original sample, models each bootstrap sample, and then combines the predictions of multiple decision trees to obtain the final prediction result through voting. A large number of theoretical and empirical studies [35–37] have proved that random forest has a high prediction accuracy, a good tolerance for outliers and noise and is not easy to overfit. In the training phase, the random forest uses bootstrap sampling to collect multiple different sub-training data sets from the input training data set to train multiple different decision trees in turn. The model architecture diagram of random forest is shown in Figure 3.

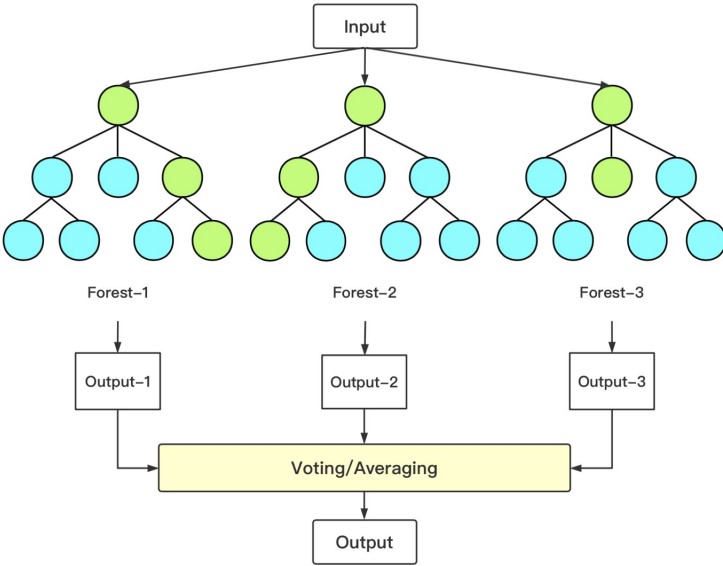

**Figure 3.** The structure of the random forest model. To return a new object from an input vector, put the input vector down each of the trees in the forest. Each tree gives a classification or a regression. The forest chooses the classification having the most votes (over all the trees in the forest), or it averages the prediction results of multiple internal decision trees.

### 3.3. MSBP Model

In this paper, the BP neural network model is modified and adjusted, and the MSBP model (Multi-step Back Propagation) is proposed. The model architecture process is shown in the Figure 4. The model is divided into four parts: site data, data processing, model, and result.

The station data are the acquired watershed data, including the rainfall data of each rainfall station in the entire watershed and the flow data of the river channel. Due to the influence of the monsoon climate in this basin, the rainfall is concentrated in June to August, which causes floods in the basin to occur more frequently in June to August. Therefore, the site data need to be processed reasonably to remove invalid data.

The data are processed to perform feature screening on the station rainfall data. There are many rainfall monitoring stations in the basin, and not all stations will have an impact on the river flow at the monitoring points. Therefore, the correlation analysis of rainfall stations can be carried out, and the rainfall stations with strong correlation can be selected as the model input data.

The core part of this model lies in the establishment of BP neural networks for simultaneous prediction to achieve multi-step prediction. Unlike the traditional BP model, which uses one model to output the results N hours in advance, this model builds N BP models, outputs the results 1, 2...N h in advance, and splices them. In parallel processing, the MSBP model inherits the automatic tuning process of the BP neural network. For the data input to the model, the hidden layer is processed, and the results are back-propagated to minimize the error loss and output the optimal results. The established many BP models can simultaneously process rainfall data in the river basin at different time scales, so that the model will output the watershed flow value for several hours in the future, and the average value can be calculated after multiple runs to obtain the final trend of future river basin changes.

The result of this model is not a single data point in the traditional BP model, but a time series. In Figure 4, T represents forecast T hours ahead, and M and N both represent the number of time windows, that is, the number of models. M can be any number from 1 to N. In addition, the T-N-M model represents any model from 1 to N models. In other words, the flood sequence we forecast does not necessarily start from 1 h and end in the next N hours, such as 1–10 h and 1–15 h. The time series may be a time series of M to N hours, for example, a time series of 6–10 h and 5–10 h.

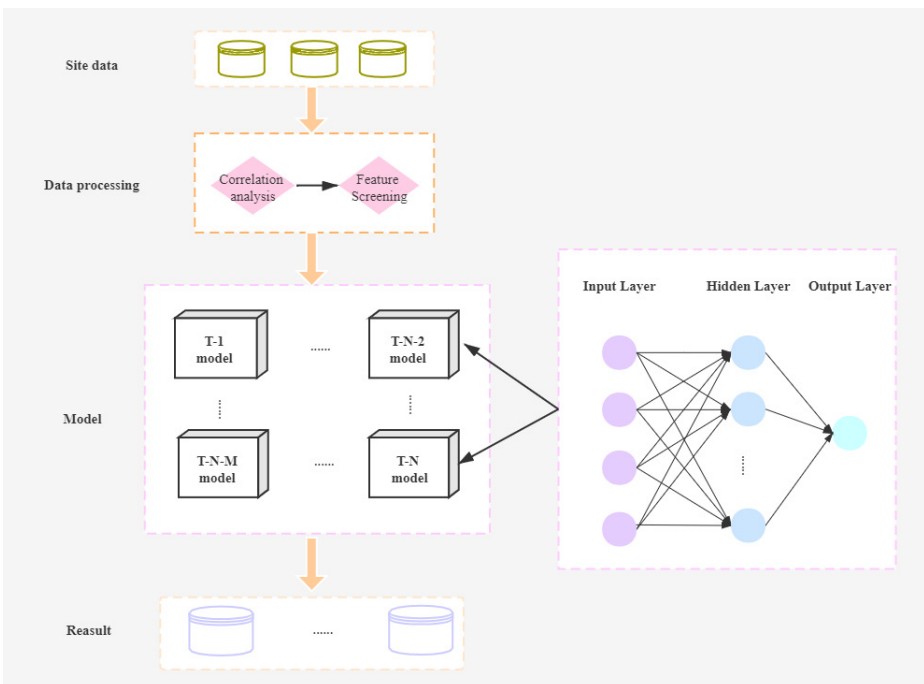

**Figure 4.** In the figure, T means forecast T hours in advance, and M and N both represent the size of time windows. When the time window is N, the number of BP neural network models is N. Among them, 0 < M < N, T-N-M > 0.

### 3.4. Evaluation Index

According to the Hydrological Information Forecast Specification (GBT 22482-2008), this model uses three evaluation indicators, namely the relative error of flood peak, the error of peak present time and the certainty coefficient to verify and analyze the forecast accuracy.

(1) Relative error of flood peak. Rainfall and runoff forecasts take 20% of the measured peak flow as the allowable error; the river flow (water level) forecast takes 20% of the measured change within the forecast period as the allowable error. When the flow allowable error is less than 5% of the actual measured value, take 5% of the actual measured value of the flow. When the allowable error of the water level is less than 5% of the measured flood peak flow, the corresponding water level amplitude value or less than 0.10 m, then this value is taken as the allowable error.

Relative error calculation formula:

$$\delta = \frac{Q_c - Q_o}{Q_o} \times 100\% \tag{1}$$

Among them, $Q_c$ represents the flow forecast value, and $Q_o$ represents the the true value of the flow.

(2) Peak present time error. The peak present time takes 30% of the time interval between the forecast based time and the actual measured flood peak as the allowable error. When the allowable error is less than 1 h or the calculation period is long, 1h or the calculation period is the allowable error.

(3) The formula for calculating the coefficient of certainty is:

$$DC = 1 - \frac{\sum_{i-1}^{n}(Q_c - Q_o)^2}{\sum_{i-1}^{n}(Q_c - \bar{Q}_o)^2} \tag{2}$$

Among them, $Q_c$ represents the flow forecast value, $Q_o$ represents the true value of the flow, $\bar{Q}_o$ represents the average value of the true value of the flow, and n represents the length of the forecast time series.

(4) When the error of a forecast is less than the allowable error, it is a qualified forecast. The percentage of the ratio of the number of qualified forecasts to the total number of forecasts is the qualified rate, which represents the overall accuracy level of multiple forecasts.The formula for calculating the qualified rate is:

$$QR = \frac{n}{m} \times 100\% \tag{3}$$

Among them, n is the number of qualified forecasts; m is the total number of forecasts.

(5) The evaluation criteria for the accuracy grade of flood forecasting results are shown in Table 1.

**Table 1.** Accuracy grades of forecast results.

| Project | | Coefficient of Certainty | Pass Rate/% |
|---------|---|--------------------------|-------------|
| | A | DC >= 0.90 | QR > 85.0 |
| acc class | B | 0.90 > DC >= 0.70 | 85.0 > QR >= 70.0 |
| | C | 0.70 > DC >= 0.50 | 70.0 > QR >= 60.0 |

(6) Nash efficiency coefficient (NSE) is generally used to verify the quality of the hydrological model simulation results. The value of NSE is negative infinity to 1, and NSE is close to 1, indicating that the model is of good quality and the model has high reliability; NSE close to 0, indicating that the simulation result is close to the average level of the observations, that is, the overall result is credible, but the process simulation error is large;

NSE is far less than 0, the model is not credible. The calculation formula of NSE is as follows:

$$NSE = 1 - \frac{\sum_{i=1}^{n}\left(Q_p^i - Q_m^i\right)^2}{\sum_{i=1}^{n}\left(Q_p^i - \bar{Q}_p\right)^2} \qquad (4)$$

Among them, $Q_p$ is the number of qualified forecasts; $Q_m$ is the total number of forecasts; $\bar{Q}_p$ represents the average value of the true value of the flow.

## 4. Results

### 4.1. Evaluation of Model Performance

The MSBP model is improved on the traditional BP neural network, and the simple single-step forecast is improved into a multi-step time series forecast. In contrast, MSBP has a better recursive structure, which can simultaneously predict traffic at different time points in the future, and the error reverse adjustment is more effective and can reach the optimum in a shorter time. Compared with BP neural network, random forest model and LSTM model, MSBP can shorten the model running speed to 5–10 s. BP neural network and random forest are single-step prediction models, and the running time is one minute. The LSTM model can realize multi-step prediction. When there is a large amount of data, the prediction time is up to 4 h. This article puts more attention on the accuracy, stability and reliability of the model. Therefore, different methods are used to perform multiple training verifications on the training set and the verification set. Each method has a different performance in the prediction results of the best peak present time.

The MSBP model can accurately predict the trend of single-peak floods or retest flood peaks. For the floods that have occurred, we numbered them according to the time of occurrence, and we used this to test the data set. For example, No. 19940805 represents the flood that occurred on 5 August 1994. This article selects No. 19940805 and No. 20170727 for training. After multiple training and averaging, the prediction results are stable, and the best results can be predicted 20 h in advance.

As shown in Figure 5, when training on No. 19940806, the effect is optimal when the prediction time is 20 h. Not only can it predict the changes in the flow in the basin, but it can also be accurate for the peak occurrence time and flood peak value. After the peak of Hong has been trained for many times, the error value is between 10 and 50 m$^3$/s, which can also be prevented in advance. The NSE reaches 0.87, and the accuracy rate is 0.9669.

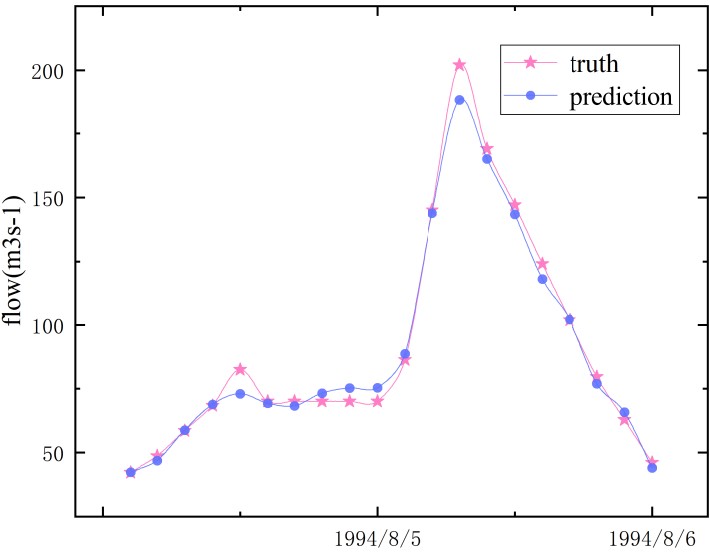

**Figure 5.** No. 19940806 MSBP model prediction. Pink is the true value, and blue is the predicted value.

Figure 6 shows the comparison of the 3 and 6 h prediction results of the MSBP model. Although it is possible to predict the change trend of the river flow, the relative error of the flood peak and the peak present time are quite different. Compared with the 20 h forecast, there is still a big gap.

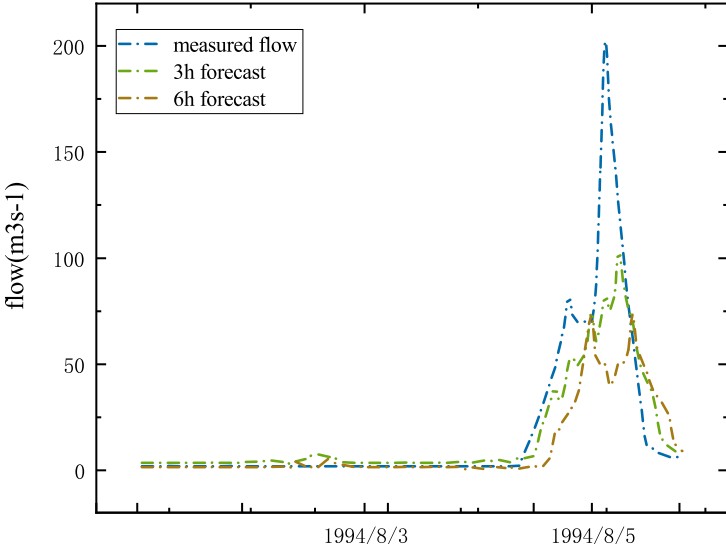

**Figure 6.** The forecasting process of the No. 19940805 flood. The blue line is the actual measured flow of the Zhaishang site; the green line is the forecast three hours in advance; the brown line is the forecast six hours in advance.

There were two peaks on the No. 20170727, and the time interval was less than 24 h. MSBP can also accurately predict this situation Figure 7. Although the prediction results of the MSBP model for the largest flood peak have a large error, the MSBP model has made more accurate predictions for the flow data of other time periods, and the peak present time prediction is more accurate. The NSE is 0.89, and the accuracy rate is 0.9564.

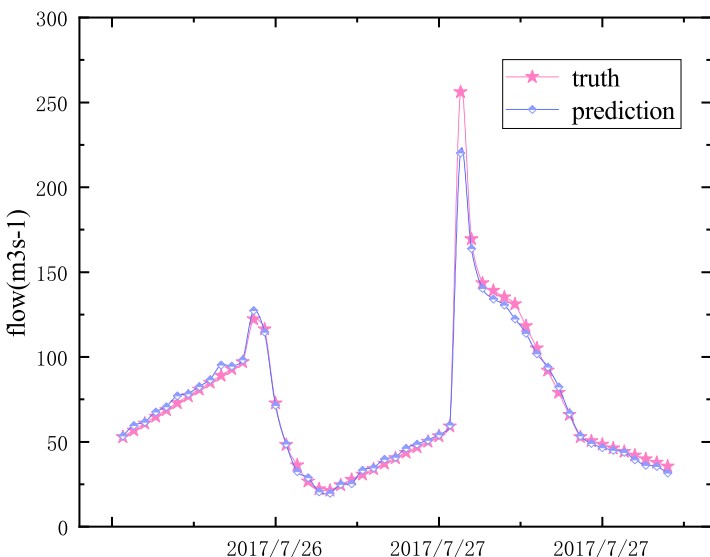

**Figure 7.** No. 20170727 MSBP model prediction. Pink is the true value, and blue is the predicted value.

Figure 8 shows the 3 h and 6 h forecasts made by the MSBP model on No. 20170727. The forecast results are poor, and the peak present time and the flood peak error are quite different. Compared with the best 20 h forecast time result, there are obvious shortcomings.

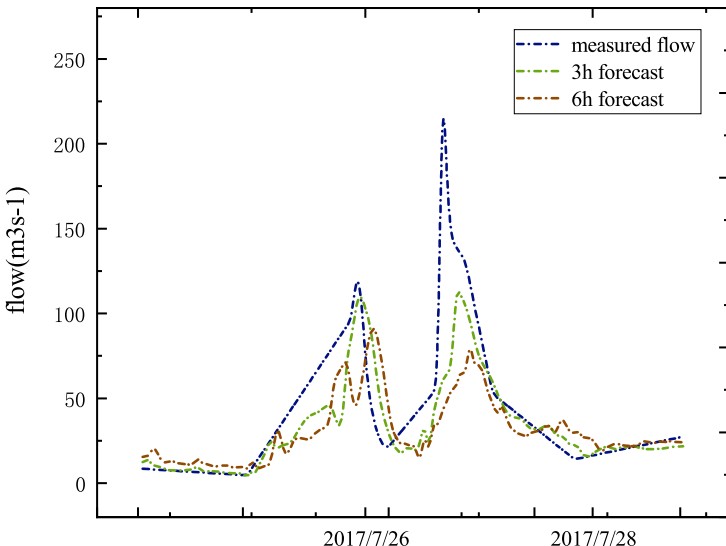

**Figure 8.** The forecasting process of the No. 20170727 flood. The blue line is the actual measured flow of the Zhaishang site; the green line is the forecast three hours in advance; the brown line is the forecast six hours in advance.

Regardless of single-peak flood or compound peak flood, the MSBP model has good prediction results. In this way, no matter how the flow in the river basin changes in the future, the MSBP model can accurately predict the flow based on the rainfall monitoring data at the stations. This also reflects the advantages of data-driven forecasting models. As long as you ensure that you have massive and accurate real data, after special preprocessing, you can get effective and accurate forecast results.

For time series forecasting, deep learning can well capture the changing trends of things over time, so as to achieve accurate time forecasts. After analyzing and comparing the data in the previous period, we found that the river has seasonal characteristics, and the flood is up to 24 h from the appearance to the end. Comparing the prediction results of the MSBP model at different times on No. 19940806 and No. 20170727, it can be seen that the MSBP model has a good effect 20 h in advance. In our opinion, forecasting 20 h in advance can take into account the entire event process from the appearance of the flood to the end, including some floods that occurred in a short period of time. However, the prediction effect of 3 h and 6 h in advance is poor because it is difficult to fully consider the flood cycle, and it only considers the changes in the part of the time when the flood occurs. Therefore, if the complete time trend can be input into the model, the MSBP model can provide more accurate prediction results.

*4.2. Model Reliability Evaluation*

In this paper, while using the MSBP model for multi-step sequence prediction, the BP neural network and random forest are used to predict the flow of a station in the basin one hour in advance. Research has shown that the quality of the data currently obtained in the basin is good, and the work carried out on the data set is effective, which confirms the reliability and practicability of the model. However, compared with the MSBP model, the single-step forecasting model, that is, the BP neural network and the random forest model, cannot perform multi-step time series forecasting, resulting in a lack of use space in practical applications.

As shown in Figure 9a, it is the BP neural network prediction result of a single peak, the No. 19940805. According to the rainfall information measured by the rainfall station, floods will occur at the Zhaishang site when large rainfall occurs in the current basin. The BP neural network model can accurately predict the occurrence of flood peaks based on rainfall information. According to "Hydrological Information Forecast Specification" (GBT 22482-2008), the relative error of No. 19940805 is 0.16, the error of peak present time

is 1 h, and the certainty coefficient is 0.9353. From this, it can be seen that the prediction level of No. 19940805 reached Class A.

As shown in Figure 9b, it is the random forest prediction result of No. 19940805. According to the "Specifications for Hydrological Information Forecasting" (GBT 22482-2008), the relative error of the No. 19940805 flood peak is 0.28, the peak present time error is 1 h, and the certainty coefficient is 0.9325. From this, it can be seen that the prediction level of No. 19940805 reached Class A. Compared with random forest, BP neural network has improved in peak value and certainty coefficient.

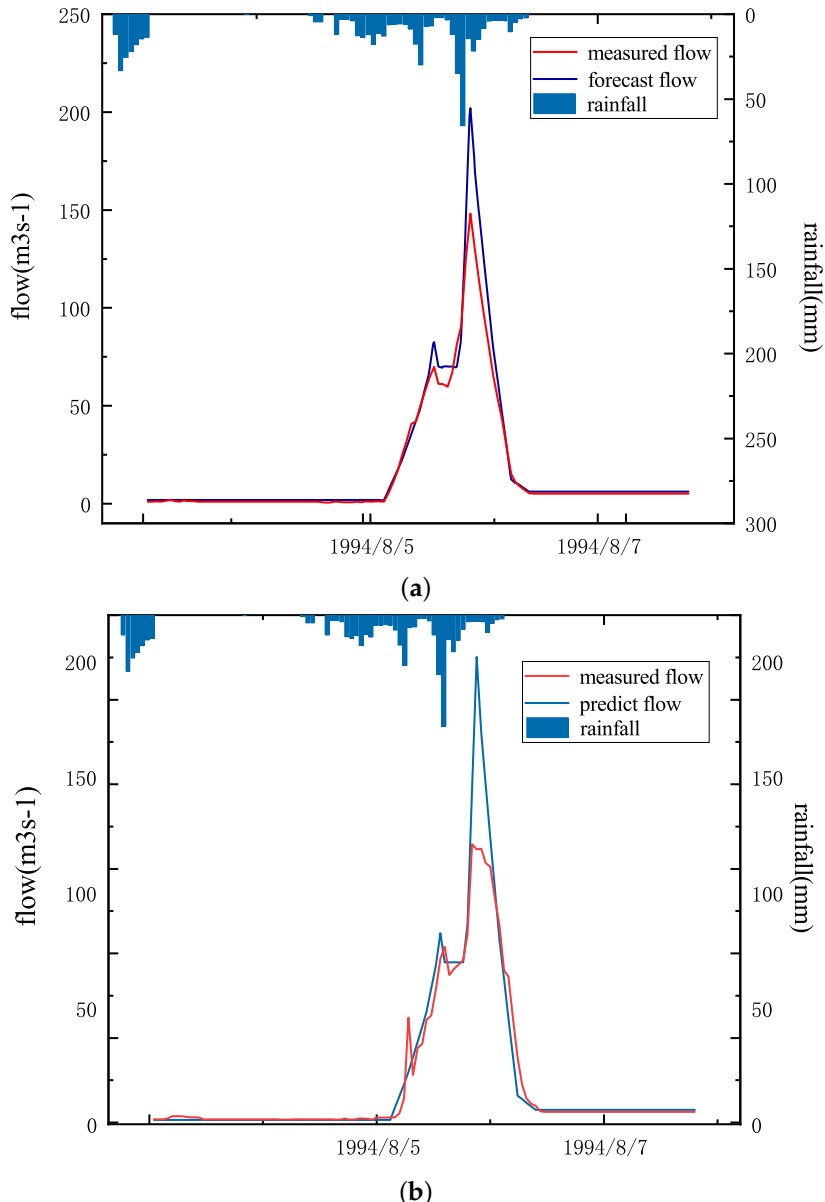

**Figure 9.** The forecasting process of the No. 19940805 flood. (**a**) Flood forecast results of No. 19940805 flood under the BP neural network model. (**b**) Flood forecast results of No. 19940805 flood under the random forest model. Among them, the blue line is the measured flow at the Zhaishang station, the red line is the predicted flow at the Zhaishang station, and the blue histogram is the rainfall value in the basin.

As shown in Figure 10a, it is the BP neural network prediction result of a compound flood peak, namely the No. 20170727 flood. According to the "Specifications for Hydrological Information Forecasting" (GBT 22482-2008), the relative error of the No. 20170727 flood

peak is 0.18, the error of the peak present time is 1 h, and the certainty coefficient is 0.9812. From this, it can be seen that the prediction level of No. 20170727 reached Grade A.

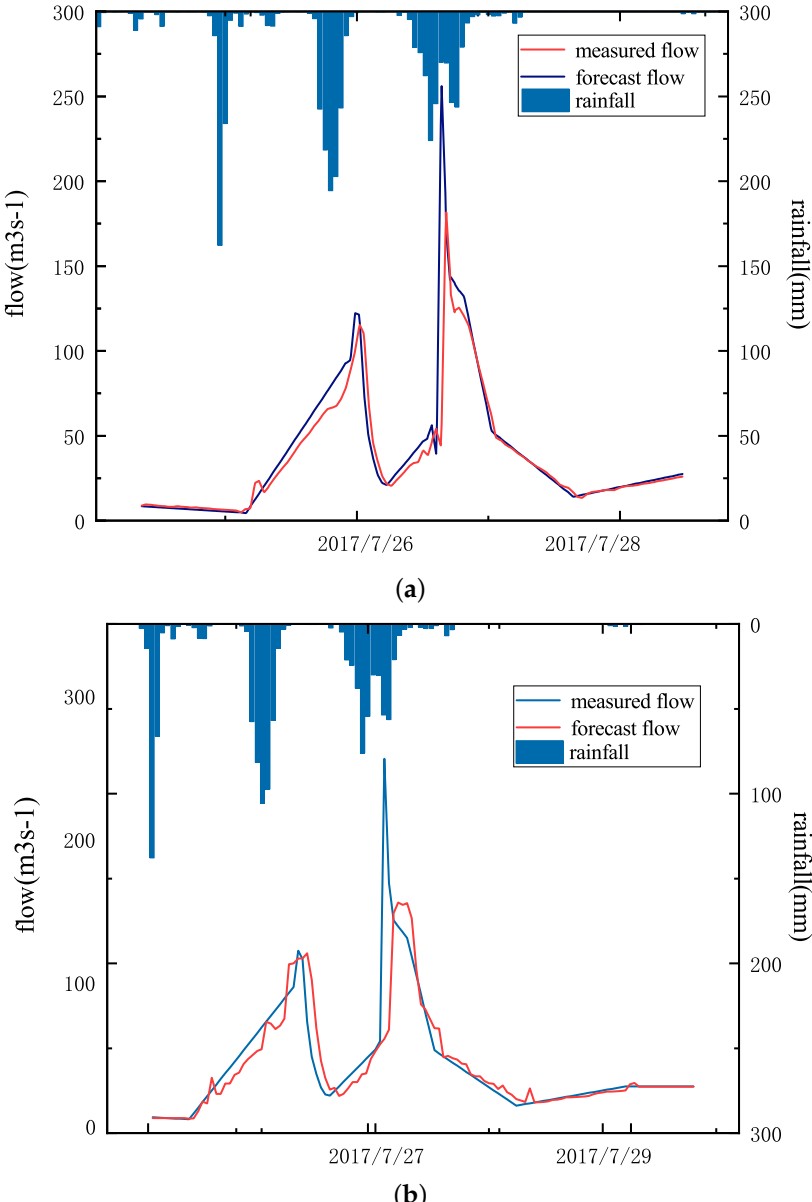

**Figure 10.** The forecasting process of flood No. 20170727. (**a**) Flood forecast results of the No. 20170727 flood under the BP neural network model. (**b**) Flood forecast results of the No. 20170727 flood under the random forest model. Among them, the blue line is the measured flow at the Zhaishang station, the red line is the predicted flow at the Zhaishang station, and the blue histogram is the rainfall value in the basin.

As shown in Figure 10b, it is the random forest prediction result of the No. 20170727. According to "Specifications for Hydrological Information Forecasting" (GBT 22482-2008), the relative error of No. 20170727 flood peak is 0.26, the error of peak present time is 1 h, and the certainty coefficient is 0.9639. From this, it can be seen that the prediction level of No. 20170727 reached Grade A. Compared with random forest, BP neural network has improved peak value and certainty coefficient.

In Figure 11, the model scatter plots of the BP neural network and random forest used in the training and testing phases are shown, respectively. The prediction results show that the two models are generally consistent with the real measurement data of the two stages.

In general, the BP neural network model has better performance than the random forest model (higher DC and NSE values, narrower discrete points). It can be concluded that the BP neural network can make more accurate predictions in the Zhaishang watershed than the random forest, and it is more suitable for the Fenhe River Basin.

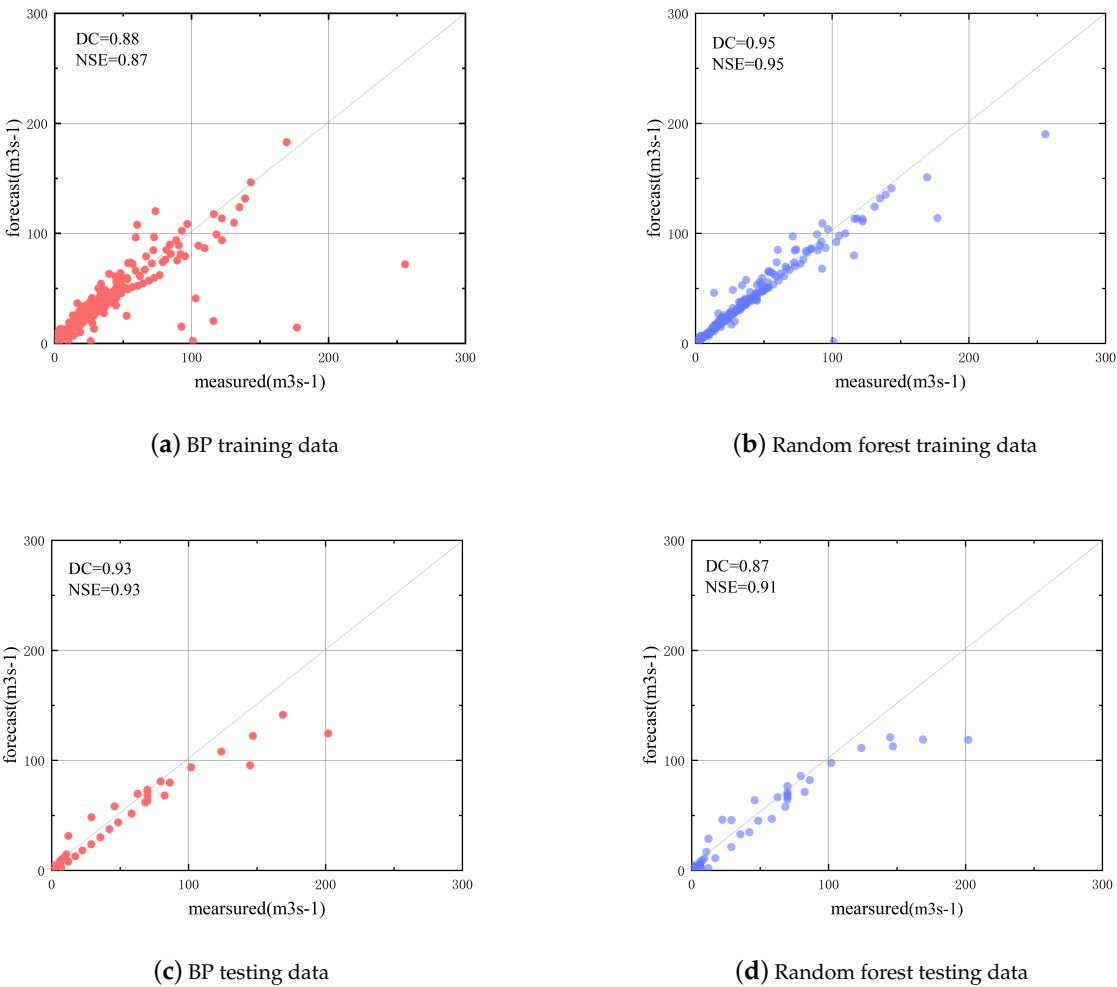

(**a**) BP training data

(**b**) Random forest training data

(**c**) BP testing data

(**d**) Random forest testing data

**Figure 11.** Scatter pictures of BP neural network and random forest for forecasting.

Based on the current rainfall data to predict the flow of the Zhaishang site in the next hour, that is, predict one hour in advance, the prediction results can reach Grade A. At the same time, the BP neural network is used to forecast floods in other time periods, and the prediction results are shown in Table 2. The relative error qualification rate of flood peak reached 81.82%, and the error of peak present time reached 100%. Taking the qualification rate and the certainty coefficient as the reference factors of the evaluation grade at the same time, the BP neural network model has reached the second-level flood prediction effect for the Fenhe Reservoir, Zhaishang Interval.

Table 3 shows the prediction results of 11 floods using random forest. Compared with the BP neural network, although the prediction level has reached the second level, the random forest forecast has reduced the relative error of the peak and the time error of the peak, and the overall certainty coefficient is not as high as the BP neural network model.

On the 11 flood data sets, BP neural network and random forest have good performance. Different single-step prediction models have complete predictions for river flow. Although part of the time is shifted back when predicting the peak present time, it can accurately predict the changing trend of the river basin. The forecast results can be used as

a reference for the staff of the Bureau of Hydrology and take preventive measures to avoid a large number of economic and personnel losses. At the same time, good research results show that the data set used in this article is of high quality, ensuring the effectiveness of this research. In addition, using the same data measurement, compilation and processing styles, other stations in the basin can get a good forecast effect in advance by using the method in this paper.

**Table 2.** Analysis of BP neural network prediction results.

| Number | Measured Flood Peak /m$^3$s$^{-1}$ | Forecast Flood/m$^3$s$^{-1}$ | Relative Error of Flood Peak | Peak Present Time Error/h | Coefficient of Certainty |
|---|---|---|---|---|---|
| 19940806 | 202 | 168.691383 | 0.165312 | 1 | 0.9353 |
| 19960803 | 101 | 101.367055 | 0.003634 | 0 | 0.9958 |
| 19990827 | 92.6 | 86.357557 | 0.064794 | 0 | 0.9945 |
| 20010810 | 103 | 107.018234 | 0.038834 | 0 | 0.9928 |
| 20060903 | 116 | 106.329911 | 0.086206 | 1 | 0.9838 |
| 20080814 | 177 | 169.988022 | 0.045192 | 0 | 0.9924 |
| 20160719 | 127 | 127.105844 | 0.000834 | 0 | 0.9946 |
| 20160824 | 116 | 118.548042 | 0.017244 | 0 | 0.9972 |
| 20170723 | 123 | 123.748795 | 0.006052 | 0 | 0.9971 |
| 20170727 | 256 | 208.145518 | 0.185937 | 0 | 0.9812 |
| 20180713 | 49 | 52.834592 | 0.061235 | 1 | 0.9468 |
| pass rate | | | 81.82% | 100% | |

**Table 3.** Analysis of random forest prediction results.

| Number | Measured Flood Peak /m$^3$s$^{-1}$ | Forecast Flood/m$^3$s$^{-1}$ | Relative Error of Flood Peak | Peak Present Time Error/h | Coefficient of Certainty |
|---|---|---|---|---|---|
| 19940806 | 202 | 144.743865 | 0.282178 | 1 | 0.9325 |
| 19960803 | 101 | 80.102079 | 0.192334 | 1 | 0.7176 |
| 19990827 | 92.6 | 91.138090 | 0.015798 | 1 | 0.6377 |
| 20010810 | 103 | 99.780720 | 0.031255 | 1 | 0.6756 |
| 20060903 | 116 | 101.211631 | 0.129301 | 1 | 0.9518 |
| 20080814 | 177 | 117.492388 | 0.338922 | 0 | 0.8848 |
| 20160719 | 127 | 120.296111 | 0.055118 | 0 | 0.9705 |
| 20160824 | 116 | 101.780022 | 0.129313 | 1 | 0.9093 |
| 20170723 | 123 | 110.284302 | 0.103379 | 2 | 0.8309 |
| 20170727 | 256 | 188.668318 | 0.265635 | 0 | 0.9639 |
| 20180713 | 49 | 54.021158 | 0.102472 | 0 | 0.9351 |
| pass rate | | | 72.73% | 90.91% | |

## 5. Conclusions

Since the current basin has a monsoon climate, the time of flood occurrence is relatively concentrated, and the current available data in the basin are lacking, so the 11 floods from 1994 to 2018 were selected for training and research. Research shows that the MSBP model proposed in this paper can not only improve the accuracy of flood forecasting but also provide new ideas for multi-step time series forecasting in the basin.

The MSBP model uses the advantage of the BP neural network that does not consider the specific connection between the input sequences. It modifies the single-step prediction to multi-step sequence prediction, and it performs feature screening, which can accurately output the optimal sequence and reduce the complexity of the traditional forecasting model. In addition, the MSBP neural network model has the advantages of convenience, quick operation, simple operation and high efficiency, which can provide a useful reference for flood forecasting in other regions with seasonal characteristics.

**Author Contributions:** Y.Z., J.R., R.W., F.F. and W.Z. designed the project. Y.Z., R.W., F.F. analyzed the data. Y.Z., R.W., F.F. and W.Z. wrote the paper. All authors have read and agreed to the published version of the manuscript.

**Funding:** This work is supported by National Natural Science Foundation of China, Grant No. 11702289, Key core technology and generic technology research and development project of Shanxi Province, No. 2020XXX013 and National Key Research and Development Project.

**Institutional Review Board Statement:** Not applicable.

**Informed Consent Statement:** Not applicable.

**Data Availability Statement:** Not applicable.

**Conflicts of Interest:** The authors declare no conflict of interest.

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
