# Peer review of "Multi-Step Sequence Flood Forecasting Based on MSBP Model"

_water, doi:10.3390/w13152095_

Round 1

Reviewer 1 Report

The proposed method and analysis sounds interesting, but unfortunately I found the description somewhat confusing and not well explained. It is not quite clear how the multistep back propagation method works an how it is related to the other discussed methods.

Maybe it would help to first introduce the BP method with its results and then show how the MSBP method improves on that and finally offer the random forest method as an alternative comparison. By the discreption now it is, for example, for a long time not clear whether the MSBP method employs a random forest somewhere.

I also did not fully understand the presented correlation analysis. My understanding is that there are over 50 locations where data is collected and only 10 of them are used. Is the shown matrix just for those, are all of the sites in the matrix used as inputs to the neural network?

In the results I am missing an interpretation on the forecasting capabilities to be better for 20 h than for less hours. Wouldn't we expect better predictions for shorter look-ahead times? How can this be explained? What are the look-ahead times for the comparisons (BP, Random Forest)?

Attached is a commented PDF with remarks and questions on details.

Author Response

回复:手稿 ID:water-1274896

手稿类型:文章

基于MSBP模型的多步序列洪水预报

     作者:张悦、任娟慧、王瑞、方飞腾、郑文

亲爱的编辑和审稿人,

我们要感谢审稿人的意见和批评,并感谢您给我们修改论文的机会。本文使用的数据集为非公开数据集,因此使用BP神经网络和随机森林来验证数据的正确性。另外,本文采用提出的MSBP模型在数据集上进行验证,上述方法均取得了较好的效果。MSBP模型可以提前20小时准确预测河流流量变化,NSE可达0.89。

According to the referees’ comments and criticisms, we have performed more simulations and analyses and made necessary changes to the manuscript. Changes to the main text of the manuscript are highlighted in red. We hope that our paper is now acceptable for publication in WATER.

Enclosed please find our response to the referees’ reports.

Sincerely yours,

Yue Zhang

Juanhui Ren

Rui Wang

Feiteng Fang

Wen Zheng

Response to Referee 1’s report

We greatly appreciate Referee 1’s positive attitude toward the publication of our paper and his/her suggestions to improve our paper.

Report from Referee 1:

The following are the revised questions and replies based on comments:

1.-The proposed method and analysis sounds interesting, but unfortunately I found the description somewhat confusing and not well explained. It is not quite clear how the multistep back propagation method works an how it is related to the other discussed methods. 

Response: A detailed introduction to the MSBP model has been added to the article, which has a more specific explanation than before. The BP neural network and random forest model listed in this article are used to make single-step predictions in the watershed to verify the feasibility of the data set. At the same time, the MSBP model proposed in this paper is an algorithm improvement carried out on the BP neural network to realize multi-step time series forecasting.

2.-Maybe it would help to first introduce the BP method with its results and then show how the MSBP method improves on that and finally offer the random forest method as an alternative comparison. By the description now it is, for example, for a long time not clear whether the MSBP method employs a random forest somewhere.

Response:This article first introduces the MSBP model, as the main content of this article,  displaying and analyzing the results of the model.The subsequent analysis of the results of BP neural network and random forest model is the auxiliary experiment of this article.A simple single-step prediction model was used to verify the operability of the screening of impact factors and the data set in the watershed.The data set is actually measured data in the watershed. So it needs to be tested by the algorithms to ensure that the data is correct and that multiple algorithms can be tested and trained on this data set.

3.-I also did not fully understand the presented correlation analysis. My understanding is that there are over 50 locations where data is collected and only 10 of them are used. Is the shown matrix just for those, are all of the sites in the matrix used as inputs to the neural network?

Response:The purpose of correlation analysis in this paper is to reduce the input parameters of the model and reduce the complexity of the model. There are currently 58 rainfall stations in the basin, and each station has data records. After the correlation analysis, the data of the 10 rainfall stations with the largest influencing factors are finally selected as input. As shown in picture 4, it is the rainfall station with the largest impact factor ,of which data is put into the model after screening. However, other sites cannot be displayed in the picture due to the large number, so they are not drawn.

4.-In the results I am missing an interpretation on the forecasting capabilities to be better for 20 h than for less hours. Wouldn't we expect better predictions for shorter look-ahead times? How can this be explained? What are the look-ahead times for the comparisons (BP, Random Forest)?

Response:The MSBP model of this article also predicts less than 20 hours. The article shows the results of forecasting 3 hours in advance and 6 hours in advance. The results show that the prediction effect is not as good as 20 hours in advance. The forecast results for the other hours are not on display.

5.-“NSE coefficient”Missing introduction of abbreviation.

Response:Modified NSE coefficient to NSE in the article.

6.-”Muti-step Back Propagation”Move to first mention of abbreviation.

Response:In the article, "Muti-step Back Propagation" has been modified to the place mentioned for the first time by MSBP.

7.-”only needs to input historical hydrological data in the basin ”It's not quite clear to me here, what the difference in the required data for the two approaches is. Maybe you could stress the difference between the two different inputs a little more.

Response:Both methods have been elaborated in the article.

8.-”has a wider application  range and stronger normalization ability ”wider and stronger than what?

Response:This compares the traditional prediction methods based on hydrological knowledge, which has a better scope of application and generalization ability, which has been explained in the article.

9.-BP”Please introduce the abbreviation.

Response:The full name of BP has been added to the text.

  1. -traditional”Does this refer to hydrological, knowledge based forecasting, or genetic algorithms / swarms?

Response:This refers to the traditional hydrological knowledge-based model, which has been explained in detail in the article.

11.-that”should this be "where" instead of "that"?

Response:I agree with your opinion and have been revised in the article.

12.-forecastin”forecasting (missing "g")

Response:Modified in the article.

13.-”above improvements based on BP neural network”Does this refer to the genetic algorithm and swarm methods mentioned above? It did not become clear from the text that those work in combination with a neural network. Other improvements are not mentioned?

Response:The improvements mentioned here are all improvements to the BP model. Compared with the models used in other cited documents, the improvements made in our model are explained in detail.

14.-UKF”Please introduce abbreviation.

Response:The full name has been added to the article.

15.-LSTM”Please introduce abbreviation.

Response:The full name has been added to the article.

16.-”gradient descent method”Could you add a reference to that method?

Response:I agree with your opinion and have added references in the article [36].

17.-”sequentially”Do you mean "subsequently"?

Response:This refers to adjusting the weights in order, from the input layer to the hidden layer, and then to the output layer.

18.-”Algorithm 1 Algorithm 1 BP neural network algorithm”Please add explanations for your symbols.

Response:An explanation has been added to the mathematical symbols included in the algorithm in the text.

19.-”Random forest”Could you add a reference to that method?

Response:I agree with your opinion and have added references [37] in the article.

20.-”studies have proved”Could you please add references to some of them?

Response:I agree with your opinion and have added references [38-40] in the article.

21.-the”Should this be "a"?

Response:I agree with your opinion and have been revised in the article.

22.-”Parallel processing on the model achieves the effect of multi-step prediction.”How does it achieve that? I am afraid that doesn't get clear.

Response:A detailed introduction has been added to the article on how the MSBP model processes data in parallel and realizes multi-step time series forecasting.

  1. -I am not completely sure, but it sounds like Q_c and Q_0 mean here something different (the peak time) than above. If so, could you please use different symbols? If they are the same could please clear up the terminology in the text?

Response:The peak time refers to the error of the peak appearance time, which is the difference calculation between the actual value and the predicted value, in minutes. Q_c and Q_0 represent the predicted value and true value of the river flow, respectively, in m^3/s, which cannot be used in the calculation of peak present time.

  1. -”compiled by the river basin.”I am not sure what is meant by that? How did the river basin compile.

Response:The original data used in this article are all recorded and archived and compiled by the Taiyuan Hydrological Bureau, and have passed the national audit standard. The specific method of compilation is confidential and cannot be informed. We only preprocessed the original data without changing its specific values. In addition, the pre-processing method was confirmed after communicating with professionals from the Taiyuan Hydrological Bureau to ensure that the method is correct. The work of this paper is based on cooperation with Taiyuan Hydrological Bureau.

  1. -”In addition, compared with BP neural network and random forest model, MSBP can shorten the model running speed to 5-10 seconds.”What are the model running times otherwise?

Response:BP neural network and random forest are single-step prediction models, and the running time is one minute. The LSTM model can realize multi-step prediction. When there is a large amount of data, the prediction time is up to 4 hours.

  1. -”No.19940805 and No.20170727”What do these numbers refer to?

Response:The above numbers are numbers based on the time of occurrence of floods that have occurred, and are used for data set testing. The numbers in the subsequent tables are the same.

  1. -”more accurate”More accurate compared to what?

Response:The comparison here is that the MSBP model has good predictions compared to the results of other algorithms mentioned in the article. The statement in the article has been modified.

28.-”which can also be prevented in advance”How can it be prevented?

Response:The work carried out in this paper is to predict the flow in the river basin. The prediction result will inform the Hydrology Bureau, and the Hydrology Bureau can take appropriate measures to prevent and avoid massive losses based on the future forecast.

29.-”The picture”Figure 8?

Response:Modified in the article.

30.-”manipulable”What do you mean with that? Why manipulate the data?

回应:我在这里要说明的是,数据集是可行的,数据被确认是正确的,对数据集进行的预处理、分析和预测是合理的,因此是可操作的。本文采用BP神经网络和随机森林模型进行训练和预测,均能取得较好的效果。可以看出,文章中使用的数据集质量不错,我们进行的多步预测工作是有效的。文章的措辞已被修改。

Reviewer 2 Report

The authors have developed a multi-step time series forecasting model based on multiple input and multiple output strategies. The so called Multi-step Back Propagation (MSBP) model was applied to the flood forecasting process of the river basin in Shanxi. Although the topic of studies is very important for the proper estimation of flood hazard in the area of examination I have a couple of remarks concerning the manuscript, first of all regarding the structure of the ms and preparation of figures. Then, the paper lacks the discussion section – the discussion with other findings and the broader discussion of results based on all 11 case studies.  Finally, I have got the feeling that it would be good if grammar and syntax of the text could be checked again by somebody fluent in English.

Specific comments are given below.

Page 1, Abstract, lines 1-2 and 8-9: poorly worded sentences

Page 1, Abstract, line 8 – after the abbreviation insert the full name of model.

Page 1, Abstract, line 9 – after the abbreviation insert the full name of coefficient.

Page 1, Abstract, lines 18, 20, 33 and many other examples – join the numbers indicating references i.e. [2,3], [5-7] or [13-16]

Page 2, line 41 – “forecasting” should be

Page 2, line 49 (UKF), line 59 (LSTM), line 64 (BP) and many other examples – explain abbreviations when appear first time in the text.

Page 2, lines 75-78 – conclusions should be in the end of ms.

Page 2, chapter 2 – before methods, the description of the dataset used in the study should be inserted. Subsections 2.1. and 2.2., describing BP Neural Network and Random Forest are written without references. Who developed these methods?

Page 2-3, Lines 91-96 - poorly worded sentences. Why “Function; The” (line 95) are written with capital letters?

Page 3, Algorithm 1 – “algorithm” is written twice, no explanation of symbols is given

Page 3, Line 103 – why “Bagging” is written with a capital letter

Page 4, Lines 118-122 – no information on data used for the analysis is given. Inform about your datasets (data period, parameters, rainfall and water level monitoring stations, floods used to test the model, etc.).

Page 4, Line – enumerate stations you used

Page 5, Figure 2 – poor quality of the figure, too small print

Page 5, lines 144-145 – who determined the allowable error? The authors or one of institutes?

Page 5-6, subsection 2.4 – some of abbreviations have different explanations:

Qc - the flow forecast value (Line 162), the predicted value of traffic (Line 154)

Qo - the true value of the flow (Line 163), the true value of traffic (Line 155) or the number of qualified forecasts (Line 182). The number of qualified forecasts may be also n (Line 170).

On the contrary for the total number of forecasts you use Qm (Line 182) or m (Line 170)

Page 6, table 1 – remove a colon in the title. Moreover explain abbreviations. The table is incomprehensible itself without further explanation. Move the table after the equation, maybe after Line 173.

Page 6, Line 169 and many other examples - number equations in the text.

Page 6, subsection 3.1. – In my view, almost no information on hydrology of examined area is given. Describe briefly the course of rainfall and river discharge during the year, historical floods, the period of examination, show the course of main rivers in the figure. The scant information in lines 196-202 is without references, period of calculations, details concerning stations.

Page 6, line 204 – what does it mean the expression “small and medium river stations”?

Page 6, line 207 – what is your period of examination?

Page 7, Figure 3 – poor quality of figure. Show geographical coordinates, scale, show names of stations, show the course of rivers and their names, increase a font size

Page 7, Figure 4 – too small print, moreover the names of stations should be given in Figure 3 to be able to follow the results.

Page 8, Line 222 – enumerate the examined 11 floods

Page 8, Line 233 – why “Data” is written with a capital letter?

Page 8, Title of chapter 4 – only “Results” sounds better

Page 8, Lines 246-249 – earlier you wrote about 11 floods tested. In the section 4.1. you mention only about two floods. Was your proposed method tested over eleven real datasets? Moreover, the spelling for floods for example No.19940805 (Line 247) is incomprehensible without an explanation. Why floods from 1994 and 2017 were chosen for the representation of results?

Page 8, Line 281 – poorly worded sentence

Pages 9-10, Figures 5, 6, 7 – correct the unit in the Y axis. m3s-1 should be.

Pages 12-12, Figures 9, 10 – correct the unit in the Y axis for m3s-1. Moreover add values next to rainfall scale (second Y axis)

Page 14, Figure 11 – too small print in subfigures

Page 14, table 2 – correct the spelling of units, explain what is written in the first column

Page 14 – add the discussion section to your ms, please. May your model be applied to other catchment areas? Discuss your results with other findings and discuss in detail your results based on all 11 case studies

Page 14, Lines 326-327 – this information should be mentioned in the materials section

Author Response

Re: Manuscript ID: water-1274896

Multi-step sequence flood forecasting based on MSBP model

     by Yue zhang, Juanhui Ren, Rui Wang, Feiteng Fang, Wen Zheng

Dear editors and referees,

We would like to thank the referees for their comments and criticisms and thank you for giving us the opportunity to revise our paper. The data set used in this article is a non-public data set, so the BP neural network and random forest are used to verify the correctness of the data. In addition, this article uses the proposed MSBP model to verify on the data set, and all the above methods have achieved good results. The MSBP model can accurately predict the change of river flow 20 hours in advance, and the NSE can reach 0.89.

According to the referees’ comments and criticisms, we have performed more simulations and analyses and made necessary changes to the manuscript. Changes to the main text of the manuscript are highlighted in red. We hope that our paper is now acceptable for publication in WATER.

Enclosed please find our response to the referees’ reports.

Sincerely yours,

Yue Zhang

Juanhui Ren

Rui Wang

Feiteng Fang

Wen Zheng

Response to Referee 2’s report

We greatly appreciate Referee 2’s positive attitude toward the publication of our paper and his/her suggestions to improve our paper.

Report from Referee 2:

The following are the revised questions and replies based on comments:

1.-Page 1, Abstract, lines 1-2 and 8-9: poorly worded sentences

Response:The sentence grammar and wording have been revised.

2.-Page 1, Abstract, line 8 – after the abbreviation insert the full name of model.

Response:The full abbreviation has been added.

3.-Page 1, Abstract, line 9 – after the abbreviation insert the full name of coefficient.

Response:The full abbreviation has been added.

4.-Page 1, Abstract, lines 18, 20, 33 and many other examples – join the numbers indicating references i.e. [2,3], [5-7] or [13-16]

Response:I agree with your opinion and have made changes in the article.

5.-Page 2, line 41 – “forecasting” should be

Response:Modifications have been made in the article.

6.-Page 2, line 49 (UKF), line 59 (LSTM), line 64 (BP) and many other examples – explain abbreviations when appear first time in the text.

Response:The full name has been added to the article.

7.-Page 2, lines 75-78 – conclusions should be in the end of ms.

Response:I agree with you. The MSBP section has put a conclusion, and the wording of this section has been revised.

8.-Page 2, chapter 2 – before methods, the description of the data set used in the study should be inserted. Subsections 2.1. and 2.2., describing BP Neural Network and Random Forest are written without references. Who developed these methods?

Response:I agree with your opinion and the article chapters have been adjusted. The BP neural network and random forest have been cited in related literature.

9.-Page 2-3, Lines 91-96 - poorly worded sentences. Why “Function; The” (line 95) are written with capital letters?

Response:The wording of the article has been revised.

10.-Page 3, Algorithm 1 – “algorithm” is written twice, no explanation of symbols is given

Response:I agree with you. The duplicate "algorithm" has been deleted, and symbol explanations have been added to the text.

11.-Page 3, Line 103 – why “Bagging” is written with a capital letter

Response:Bagging (Bootstrap aggregating) is a professional vocabulary and a group learning algorithm in the field of machine learning.

12.-Page 4, Lines 118-122 – no information on data used for the analysis is given. Inform about your datasets (data period, parameters, rainfall and water level monitoring stations, floods used to test the model, etc.).

Response:The article has a description of the data set. In section 2.1, there are river basin conditions and the sites included, and an illustration is attached.

13.-Page 4, Line – enumerate stations you used

Response:The site information we use has been described in section 2.2. After screening by impact factors, only 10 sites data are selected as input to predict the traffic data of Zhaishang sites.

14.-Page 5, Figure 2 – poor quality of the figure, too small print

Response:The picture has been optimized to increase the clarity and adjust the font size.

15.-Page 5, lines 144-145 – who determined the allowable error? The authors or one of institutes?

Response:The standards we use are from the “Hydrological Information Forecast Specification”(GBT 22482-2008), which have been explained in section 3.4 of the text.

  1. -Page 5-6, subsection 2.4 – some of abbreviations have different explanations:

Qc - the flow forecast value (Line 162), the predicted value of traffic (Line 154)

Qo - the true value of the flow (Line 163), the true value of traffic (Line 155) or the number of qualified forecasts (Line 182). The number of qualified forecasts may be also n (Line 170).

On the contrary for the total number of forecasts you use Qm (Line 182) or m (Line 170)

Response:I agree with your opinion. The text has been revised and the interpretation of symbols has been unified. Symbols with different meanings have changed their representation methods.

  1. -Page 6, table 1 – remove a colon in the title. Moreover explain abbreviations. The table is incomprehensible itself without further explanation. Move the table after the equation, maybe after Line 173.

Response:I agree with your opinion. I have made changes in the article and adjusted the position of the table.

18.-Page 6, Line 169 and many other examples - number equations in the text.

Response:All formulas in the text have been numbered.

19.-Page 6, subsection 3.1. – In my view, almost no information on hydrology of examined area is given. Describe briefly the course of rainfall and river discharge during the year, historical floods, the period of examination, show the course of main rivers in the figure. The scant information in lines 196-202 is without references, period of calculations, details concerning stations.

Response:The data set used in this article is a non-public data set. Except for the description of the river channel information, the rest of the information cannot be made public due to the particularity of the data set. In the Fenhe River Basin, data is monitored and recorded throughout the year. However, due to the seasonal characteristics of the river basin, the annual data only has reference value from June to August, which has been explained in section 3.3 of the article.

20.-Page 6, line 204 – what does it mean the expression “small and medium river stations”?

Response:According to Chinese standards, the river protection area is less than 300,000 acres, which is a small and medium-sized river. "Small and medium river stations" refer to monitoring stations established on these rivers.

21..-Page 6, line 207 – what is your period of examination?

Response:After establishing the model, we tested all the 11 floods mentioned in the article. Choose a single-peak flood and a double-peak flood for discussion to show that no matter what the flood trend is, the model can predict well.

22.-Page 7, Figure 3 – poor quality of figure. Show geographical coordinates, scale, show names of stations, show the course of rivers and their names, increase a font size

Response:The picture has been modified.

23.-Page 7, Figure 4 – too small print, moreover the names of stations should be given in Figure 3 to be able to follow the results.

Response::The picture has been modified.

24.-Page 8, Line 222 – enumerate the examined 11 floods

Response:I agree with your opinion. I have made changes in the article.

25.-Page 8, Line 233 – why “Data” is written with a capital letter?

Response:I agree with your opinion. I have made changes in the article.

26.-Page 8, Title of chapter 4 – only “Results” sounds better

Response:I agree with your opinion. I have made changes in the article.

27.-Page 8, Lines 246-249 – earlier you wrote about 11 floods tested. In the section 4.1. you mention only about two floods. Was your proposed method tested over eleven real datasets? Moreover, the spelling for floods for example No.19940805 (Line 247) is incomprehensible without an explanation. Why floods from 1994 and 2017 were chosen for the representation of results?

Response:This article tested all 11 floods. No.19940805 is a numbered representation based on the time when the flood occurred. The two floods shown in the article are single peak floods and compound peak floods, which are representative. Two different types of floods are used to show the results, indicating that the model has good flood prediction results.

28.-Page 8, Line 281 – poorly worded sentence

Response:I agree with your opinion. I have made changes in the article.

29.-Pages 9-10, Figures 5, 6, 7 – correct the unit in the Y axis. m3s-1 should be.

Response:The picture has been modified.

30.-Pages 12-12, Figures 9, 10 – correct the unit in the Y axis for m3s-1. Moreover add values next to rainfall scale (second Y axis)

Response:The picture has been modified.

31.-Page 14, Figure 11 – too small print in subfigures

Response:The picture has been modified.

32.-Page 14, table 2 – correct the spelling of units, explain what is written in the first column

Response:The first column is the flood name, which indicates the flood number according to the time when the flood occurred.

33.-Page 14 – add the discussion section to your ms, please. May your model be applied to other catchment areas? Discuss your results with other findings and discuss in detail your results based on all 11 case studies.

Response:A discussion has been added to the MSBP model section. This method can be used in other watersheds. It is currently being tested at other sites in the basin.

34.-Lines 326-327 – this information should be mentioned in the materials section

Response:This information has been explained in section 3.3 of the article.

Round 2

Reviewer 1 Report

Thanks for the revisions and added explanations. There is still some confusion, which I fear, I failed to communicate better in my previous comments.

You write: "As shown in Figure 2, the correlation analysis of 58 rainfall stations in the Fenhe Reservoir-Zhaishang Interval Basin was conducted, and finally the 10 upstream rainfall stations with the strongest correlation with the Zhaishang station were selected as the basic data, reducing the input sequence of the model and reducing the data complexity."

This sounds like we'd expect to see all 58 rainfall stations in Figure 2. However, it looks like it only shows the ten upstream rainfall stations with the strongest correlation? So figure 58 does not help to illustrate the selection at all, because the ten shown ones are the ones that have been selected. The caption of the figure states: "The figure shows that the correlation of the predictors used in this model is strongly related to the flow of Zhaishang." But it is hard for the reader to see this without context. The smallest correlation coefficient of the selected 10 sites seems to be 0.35. How large are the correlation coefficients for the other 48 sites? Why do you pick 10 sites and not use some threshold for correlation coefficient to select sites to include? What additional information is offered to the reader by the full correlation matrix between all those selected 10 sites that helps to understand the points you are making? What does the "flownow" entry in the matrix indicate?

In Figure 4 and the explanations on the MSBP approach, the terminology still escapes me. What is "T" and "M"? The explanation itself seems simple enough there are N models to predict the N-hour look-ahead, and the system can provide predictions for each hour up to the Nth. But I can't fit in T and M in there. In the Text only M is mentioned: "Therefore, the prediction result can be not only x hours ahead, x  [1, N], but also y hours ahead, y  [M, N],1<M<N. Which makes it look like M is an index, not some fixed number? What T is supposed to be is not explained at all as far as I can see. I'd guess it is the maximum look-ahead time, but wouldn't that be given by N already? And why would we then step further into the past with T-N-M? That would imply we use N+M models in my understanding. Could you please clarify the explanations there and indicate what the respective symbols you use mean?

In the results section, you reply above: "BP neural network and random forest are single-step prediction models, and the running time is one minute. The LSTM model can realize multi-step prediction. When there is a large amount of data, the prediction time is up to 4 hours." Could you include some indication of this information in the article? Otherwise the reader can't really assess what is meant by "shorten the model running speed to 5-10 seconds" due to lack of comparison.

You explain in your reply: "The above numbers are numbers based on the time of occurrence of floods that have occurred, and are used for data set testing. The numbers in the subsequent tables are the same." Could you please add a reference to the table at that point or maybe point out the database these numbers refer to?

With respect to the results you state in your reply: "The MSBP model of this article also predicts less than 20 hours. The article shows the results of forecasting 3 hours in advance and 6 hours in advance. The results show that the prediction effect is not as good as 20 hours in advance. The forecast results for the other hours are not on display."

This doesn't address my question. The paper is clear enough on the explanation that all the hours up to the maximum are computed by the method. But what is a little bit surprising is, that the prediction further into the future (20h) is better than the prediction into earlier points in time (3 and 6 hours shown). We'd usually expect better predictions for shorter look-ahead times, wouldn't we? You write that 20h is optimal, did you also do predictions for longer look-ahead times (what was the N used in your computations)? Do you have any explanation for this interesting observation?

It still is unclear to me which look-ahead times were used in the BP and Random Forest comparison simulations. Are these also 20h look-aheads? As the prediction seems to depend so much on this look-ahead time that would be an important information.

Author Response

Re: Manuscript ID: water-1274896

Multi-step sequence flood forecasting based on MSBP model

     by Yue zhang, Juanhui Ren, Rui Wang, Feiteng Fang, Wen Zheng

Dear editors and referees,

We would like to thank the referees for their comments and criticisms and thank you for giving us the opportunity to revise our paper. The data set used in this article is a non-public data set, so the BP neural network and random forest are used to verify the correctness of the data. In addition, this article uses the proposed MSBP model to verify on the data set, and all the above methods have achieved good results.

According to the referees’ comments and criticisms, we have performed more simulations and analyses and made necessary changes to the manuscript. Changes to the main text of the manuscript are highlighted in red. We hope that our paper is now acceptable for publication in WATER.

Enclosed please find our response to the referees’ reports.

Sincerely yours,

Yue Zhang

Juanhui Ren

Rui Wang

Feiteng Fang

Wen Zheng

Response to Referee 1’s report

We greatly appreciate Referee 1’s positive attitude toward the publication of our paper and his/her suggestions to improve our paper.

Report from Referee 1:

The following are the revised questions and replies based on comments:

  1. -You write: "As shown in Figure 2, the correlation analysis of 58 rainfall stations in the Fenhe Reservoir-Zhaishang Interval Basin was conducted, and finally the 10 upstream rainfall stations with the strongest correlation with the Zhaishang station were selected as the basic data, reducing the input sequence of the model and reducing the data complexity."

This sounds like we'd expect to see all 58 rainfall stations in Figure 2. However, it looks like it only shows the ten upstream rainfall stations with the strongest correlation? So figure 58 does not help to illustrate the selection at all, because the ten shown ones are the ones that have been selected. The caption of the figure states: "The figure shows that the correlation of the predictors used in this model is strongly related to the flow of Zhaishang." But it is hard for the reader to see this without context. The smallest correlation coefficient of the selected 10 sites seems to be 0.35. How large are the correlation coefficients for the other 48 sites? Why do you pick 10 sites and not use some threshold for correlation coefficient to select sites to include? What additional information is offered to the reader by the full correlation matrix between all those selected 10 sites that helps to understand the points you are making?

Response: I am glad to answer your questions. Figure 2 shows the correlation between our filtered rainfall sites and Zhaishang flow. The analysis of the correlation between the other 48 sites and the Zhaishang site is due to the small value, which is 0.1 and below, and the data in the figure is too large to display and is not shown. At the same time, some of the 58 rainfall stations were constructed later, resulting in incomplete data. The picture below shows the information of all 58 stations in the basin. Only the locations of the stations we have selected are included in the text. There are many rainfall monitoring stations in the basin, but not all stations will have a great impact on the river flow at the Zhaishang station. Our impact factor analysis method can effectively remove sites with small impact factors in the data, so that errors caused by data redundancy can be avoided in the neural network, and the effect can be improved. The view that reducing model input variables will improve the accuracy of the model has been confirmed by other researchers, such as literature [10].

  1. -What does the "flownow" entry in the matrix indicate?

Response: Flownow represents the flow data of the Zhaishang site. At the same time, I have added a description to the article.

  1. -In Figure 4 and the explanations on the MSBP approach, the terminology still escapes me. What is "T" and "M"? The explanation itself seems simple enough there are N models to predict the N-hour look-ahead, and the system can provide predictions for each hour up to the Nth. But I can't fit in T and M in there. In the Text only M is mentioned: "Therefore, the prediction result can be not only x hours ahead, x  [1, N], but also y hours ahead, y  [M, N],1<M<N. Which makes it look like M is an index, not some fixed number? What T is supposed to be is not explained at all as far as I can see. I'd guess it is the maximum look-ahead time, but wouldn't that be given by N already? And why would we then step further into the past with T-N-M? That would imply we use N+M models in my understanding. Could you please clarify the explanations there and indicate what the respective symbols you use mean?

Response: I'm sorry that the previous explanation did not make you understand the MSBP model method. In Figure 4, T represents forecast T hours ahead, and M and N both represent the number of time windows, that is, the number of models. Where M can be any number from 1 to N. In other words, the flood sequence we forecast does not necessarily start from 1 hour and end in the next N hours like 1-10 hours and 1-15 hours. The time series may be a time series of M to N hours, for example, a time series of 6-10 hours and 5-11 hours. At the same time, I also made changes in the text.

  1. -In the results section, you reply above: "BP neural network and random forest are single-step prediction models, and the running time is one minute. The LSTM model can realize multi-step prediction. When there is a large amount of data, the prediction time is up to 4 hours." Could you include some indication of this information in the article? Otherwise the reader can't really assess what is meant by "shorten the model running speed to 5-10 seconds" due to lack of comparison.

Response: Regarding your comments, I have revised the article and added a comparative explanation in the article.

  1. -You explain in your reply: "The above numbers are numbers based on the time of occurrence of floods that have occurred, and are used for data set testing. The numbers in the subsequent tables are the same." Could you please add a reference to the table at that point or maybe point out the database these numbers refer to?

Response: I'm glad you provide suggestions for data annotations in this article. The data set in this article is a non-public data set. The numbers in the table are named by the time when the flood occurred. The meaning of these numbers has been explained in the text.

  1. -With respect to the results you state in your reply: "The MSBP model of this article also predicts less than 20 hours. The article shows the results of forecasting 3 hours in advance and 6 hours in advance. The results show that the prediction effect is not as good as 20 hours in advance. The forecast results for the other hours are not on display."

This doesn't address my question. The paper is clear enough on the explanation that all the hours up to the maximum are computed by the method. But what is a little bit surprising is, that the prediction further into the future (20h) is better than the prediction into earlier points in time (3 and 6 hours shown). We'd usually expect better predictions for shorter look-ahead times, wouldn't we? You write that 20h is optimal, did you also do predictions for longer look-ahead times (what was the N used in your computations)? Do you have any explanation for this interesting observation?

Response: Thank you for your question. Due to the special location of Shanxi Province, the rivers have seasonal characteristics, so the flood-prone period is from June to August every year. However, a large amount of data shows that the maximum time from the occurrence to the end of the flood is 24 hours. Therefore, in the study, the prediction time was trained up to 24 hours. The neural network has a good effect on the learning of the entire trend over time.We believe that the reason for the better results for the next 20 hours is that it can consider the entire event process from the appearance of the flood to the end, including some floods with a shorter time. However, the prediction effect of 3 hours and 6 hours in advance is poor because it is difficult to fully consider the flood cycle and only consider the part of the time when the flood occurs.

  1. -It still is unclear to me which look-ahead times were used in the BP and Random Forest comparison simulations. Are these also 20h look-aheads? As the prediction seems to depend so much on this look-ahead time that would be an important information.

Response: For BP neural network and random forest, we only made predictions 1 hour in advance. For our proposed multi-step time series forecasting model MSBP, because BP neural network and random forest are single-step time forecasting, they search for the optimal forecast time without great reference. We use the single-step prediction model to verify the availability of the data set, not to verify the optimal prediction time.

Reviewer 2 Report

The authors have developed a multi-step time series forecasting model based on multiple input and multiple output strategies. The so called Multi-step Back Propagation (MSBP) model was applied to the flood forecasting process of a river basin in Shanxi. The authors have rewritten the ms according to many of my suggestions. Nevertheless, I have still a few remarks to the prepared ms.  First remark regards Figure 1 (Page 3). The subfigure in the left upper corner is entirely unreadable – too small print, the subfigure in the left bottom corner - no geographic coordinates, no geographic names, no information what it is presented within it. Moreover in Figures 5 to 10 correct the unit in the Y axis: m3s-1 should be. My last remark regard section 2.1. (Lines 93-99). Please present a brief general information on the annual course of precipitation and river discharge within the examined area. In which months of the year maximum values of rainfall and river discharge are usually observed?

Author Response

Re: Manuscript ID: water-1274896

Multi-step sequence flood forecasting based on MSBP model

     by Yue zhang, Juanhui Ren, Rui Wang, Feiteng Fang, Wen Zheng

Dear editors and referees,

We would like to thank the referees for their comments and criticisms and thank you for giving us the opportunity to revise our paper. The data set used in this article is a non-public data set, so the BP neural network and random forest are used to verify the correctness of the data. In addition, this article uses the proposed MSBP model to verify on the data set, and all the above methods have achieved good results.

According to the referees’ comments and criticisms, we have performed more simulations and analyses and made necessary changes to the manuscript. Changes to the main text of the manuscript are highlighted in red. We hope that our paper is now acceptable for publication in WATER.

Enclosed please find our response to the referees’ reports.

Sincerely yours,

Yue Zhang

Juanhui Ren

Rui Wang

Feiteng Fang

Wen Zheng

Response to Referee 2’s report

We greatly appreciate Referee 1’s positive attitude toward the publication of our paper and his/her suggestions to improve our paper.

Report from Referee 2:

The following are the revised questions and replies based on comments:

  1. -First remark regards Figure 1 (Page 3). The subfigure in the left upper corner is entirely unreadable – too small print, the subfigure in the left bottom corner - no geographic coordinates, no geographic names, no information what it is presented within it.

Response: I'm glad you provide suggestions in this article. I have modified the picture based on the comments you provided.

  1. -Moreover in Figures 5 to 10 correct the unit in the Y axis: m3s-1 should be.

Response: I'm glad you provide suggestions in this article. I have modified the picture based on the comments you provided.

  1. -My last remark regard section 2.1. (Lines 93-99). Please present a brief general information on the annual course of precipitation and river discharge within the examined area. In which months of the year maximum values of rainfall and river discharge are usually observed?

Response: I'm glad you provide suggestions in this article. I have modified the article based on the comments you provided: In the Fen River Basin, due to the seasonality of the river, the rainy season is concentrated in June to August each year. Most of the floods occur from the end of July to mid-August each year.

Round 3

Reviewer 1 Report

Dear authors, thanks for the explanations and for the updated text addressing some of my questions. Some questions only seem to be answered to me, however, the reason for my questions was that, I think these kind of information are missing in the presentation of your work and should be included to the benefit of the readers.

You write in your answer to my question: "The analysis of the correlation between the other 48 sites and the Zhaishang site is due to the small value, which is 0.1 and below"

This information would be of interest to put the shown correlation data for the selected 10 sites into perspective. Because Figure 2 only shows the 10 selected sites and their correlation, it does not show that "the correlation analysis of 58 rainfall stations in the Fenhe Reservoir-Zhaishang Interval Basin was conducted", please consider rewording the explanation on the figure and the inclusion of the information for the correlation of the other sites as provided in your answer above. Were those 10 sites selected because of a threshold somewhere between 0.1 and 0.35, or did you pick the best fitting 10 sites (was of sites to include a-priori)? If you fixed the number of sites to use beforehand, why did you pick 10 and not 12 or 8? It would be kind to either include the reasoning for the specific number of sites to use or the the threshold value for the correlation coefficients.

Thanks for the explanations on the symbol M in the MSBP method. My understanding is that M>1 incurs additional models and the need to consider longer time ranges, because you have those going back to T-M-N, is that correct? Could you kindly consider making this a little more explicit in your text?

With respect to the dataset, my request was maybe not clearly enough formulated. I understand that the dataset is not public, but what I was asking for is putting a reference to the tables you include where you mention the dataset numbers to help the readers to draw the connection (add the reference to table 2 and 3 where the data numbers are introduced). It's not necessary, but I believe it could help to provide the link between the two for the reader.

Could you kindly include your explanation on the reason why the prediction is best for 20 hours in the article? I have to admit that I am still a little confused: Wouldn't an easy remedy to this worsened short term prediction be to widen the window of past data to be used in the model?

Could you please include this information "For BP neural network and random forest, we only made predictions 1 hour in advance." in the article?

Author Response

Re: Manuscript ID: water-1274896

Multi-step sequence flood forecasting based on MSBP model

     by Yue zhang, Juanhui Ren, Rui Wang, Feiteng Fang, Wen Zheng

Dear editors and referees,

We would like to thank the referees for their comments and criticisms and thank you for giving us the opportunity to revise our paper. The data set used in this article is a non-public data set, so the BP neural network and random forest are used to verify the correctness of the data. In addition, this article uses the proposed MSBP model to verify on the data set, and all the above methods have achieved good results.

According to the referees’ comments and criticisms, we have performed more simulations and analyses and made necessary changes to the manuscript. Changes to the main text of the manuscript are highlighted in red. We hope that our paper is now acceptable for publication in WATER.

Enclosed please find our response to the referees’ reports.

Sincerely yours,

Yue Zhang

Juanhui Ren

Rui Wang

Feiteng Fang

Wen Zhen

Response to Referee 1’s report

We greatly appreciate Referee 1’s positive attitude toward the publication of our paper and his/her suggestions to improve our paper.

Report from Referee 1:

The following are the revised questions and replies based on comments:

1.-This information would be of interest to put the shown correlation data for the selected 10 sites into perspective. Because Figure 2 only shows the 10 selected sites and their correlation, it does not show that "the correlation analysis of 58 rainfall stations in the Fenhe Reservoir-Zhaishang Interval Basin was conducted", please consider rewording the explanation on the figure and the inclusion of the information for the correlation of the other sites as provided in your answer above. Were those 10 sites selected because of a threshold somewhere between 0.1 and 0.35, or did you pick the best fitting 10 sites (was of sites to include a-priori)? If you fixed the number of sites to use beforehand, why did you pick 10 and not 12 or 8? It would be kind to either include the reasoning for the specific number of sites to use or the the threshold value for the correlation coefficients.

Response:Thank you for your comments, I have made changes in the paper, specifically in the first paragraph of section 2.2 (lines 108-115). At the same time, we selected 10 sites for two reasons: one is that the data of these 10 sites is relatively complete, which avoids the existence of model errors caused by abnormal data during data processing; the other is that 10 rainfall stations has much higher correlation(correlation greater than 0.1) than other stations as the basic data of the model.

2.-Thanks for the explanations on the symbol M in the MSBP method. My understanding is that M>1 incurs additional models and the need to consider longer time ranges, because you have those going back to T-M-N, is that correct? Could you kindly consider making this a little more explicit in your text?

Response:Thank you for your comments. I have made changes in the paper, specifically in section 3.3 (lines 224-229).

3.-With respect to the dataset, my request was maybe not clearly enough formulated. I understand that the dataset is not public, but what I was asking for is putting a reference to the tables you include where you mention the dataset numbers to help the readers to draw the connection (add the reference to table 2 and 3 where the data numbers are introduced). It's not necessary, but I believe it could help to provide the link between the two for the reader.

Response:Thank you for your comments, I have made changes in the paper, specifically in section 2.2 (lines 134-139).

4.-Could you kindly include your explanation on the reason why the prediction is best for 20 hours in the article? I have to admit that I am still a little confused: Wouldn't an easy remedy to this worsened short term prediction be to widen the window of past data to be used in the model?

Response:Thank you for your comments. I have made changes in the text, specifically in section 4.1 (lines 317-328). Indeed, for short-term forecasts, entering a larger time window can obtain good forecast results. We have also made other time forecasts for the current basin for more than 20 hours, but the forecasting effect has not improved very well. We consider this because the current seasonal characteristics of the river and the maximum occurrence period is 24 hours.

5.-Could you please include this information "For BP neural network and random forest, we only made predictions 1 hour in advance." in the article。

Response:Thank you for your comments. I have made changes in the paper, specifically in section 4.2 (lines 330-332).
